# Structured Object-Aware Physics Prediction for Video Modeling and Planning

**Jannik Kossen**[*1], **Karl Stelzner**[*2], **Marcel Hussing**[3], **Claas Voelcker**[3] & **Kristian Kersting**[2]
[1]Department of Physics and Astronomy, Heidelberg University
[1]`kossen@stud.uni-heidelberg.de`
[2,3]Department of Computer Science, TU Darmstadt
[2]`{stelzner,kersting}@cs.tu-darmstadt.de`
[3]`{marcel.hussing,c.voelcker}@stud.tu-darmstadt.de`

## Abstract

When humans observe a physical system, they can easily locate objects, understand their interactions, and anticipate future behavior. For computers, however, learning such models from videos in an unsupervised fashion is an unsolved research problem. In this paper, we present STOVE, a novel state-space model for videos, which explicitly reasons about objects and their positions, velocities, and interactions. It is constructed by combining an image model and a dynamics model in compositional manner and improves on previous work by reusing the dynamics model for inference, accelerating and regularizing training. STOVE predicts videos with convincing physical behavior over thousands of timesteps, outperforms previous unsupervised models, and even approaches the performance of supervised baselines. We further demonstrate the strength of our model as a simulator for sample efficient model-based control in a task with heavily interacting objects.

## 1 Introduction

Obtaining structured knowledge about the world from unstructured, noisy sensory input is a key challenge in artificial intelligence. Of particular interest is the problem of identifying objects from visual input and understanding their interactions. One longstanding approach to this is the idea of *vision as inverse graphics* (Grenander, 1976), which postulates a data generating graphics process and phrases vision as posterior inference in the induced distribution. Despite its intuitive appeal, vision as inference has remained largely intractable in practice due to the high-dimensional and multimodal nature of the inference problem. Recently, however, probabilistic models based on deep neural networks have made promising advances in this area. By composing conditional distributions parameterized by neural networks, highly expressive yet structured models have been built. At the same time, advances in general approximate inference, particularly variational techniques, have put the inference problem for these models within reach (Zhang et al., 2019).

Based on these advances, a number of probabilistic models for unsupervised scene understanding in single images have recently been proposed. The structured nature of approaches such as AIR (Eslami et al., 2016), MONet (Burgess et al., 2019), or IODINE (Greff et al., 2019) provides two key advantages over unstructured image models such as variational autoencoders (Kingma & Welling, 2014) or generative adversarial networks (Goodfellow et al., 2014). First, it allows for the specification of inductive biases, such as spatial consistency of objects, which constrain the model and act as regularization. Second, it enables the use of semantically meaningful latent variables, such as object positions, which may be used for downstream reasoning tasks.

Building such a structured model for videos instead of individual images is the natural next challenge. Not only could such a model be used in more complex domains, such as reinforcement learning, but the additional redundancy in the data can even simplify and regularize the object detection problem (Kosiorek et al., 2018). To this end, the notion of temporal consistency may be leveraged

---

[*]Both authors contributed equally to this work.

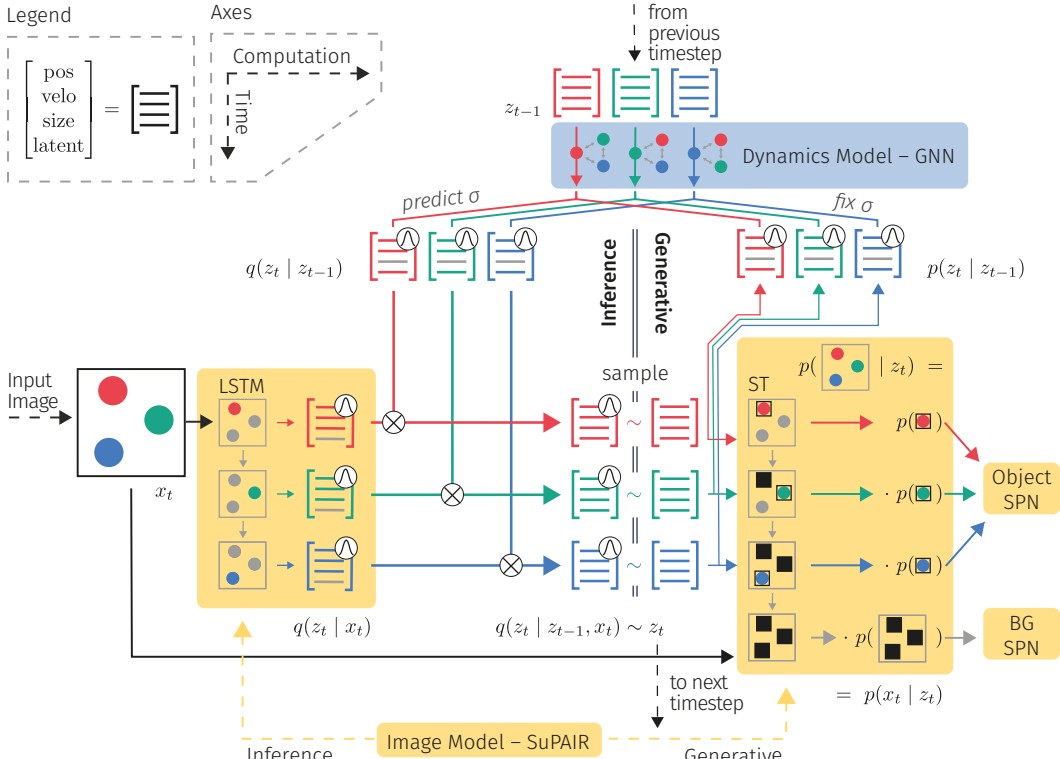

Figure 1: Overview of STOVE's architecture. (Center left) At time $t$, the input image $x_t$ is processed by an LSTM in order to obtain a proposal distribution over object states $q(z_t \mid x_t)$. (Top) A separate proposal $q(z_t \mid z_{t-1})$ is obtained by propagating the previous state $z_{t-1}$ using the dynamics model. (Center) The multiplication of both proposal distributions yields the final variational distribution $q(z_t \mid z_{t-1}, x_t)$. (Right) We sample $z_t$ from this distribution to evaluate the generative distribution $p(z_t \mid z_{t-1})p(x_t \mid z_t)$, where $p(z_t \mid z_{t-1})$ shares means – but not variances – with $q(z_t \mid z_{t-1})$, and $p(x_t \mid z_t)$ can be obtained by direct evaluation of $x_t$ in the sum-product networks. Not shown is the dependence on $x_{t-1}$ in the inference routine which allows for the inference of velocities. (Best viewed in color.)

as an additional inductive bias, guiding the model to desirable behavior. In situations where interactions between objects are prevalent, understanding and explicitly modeling these interactions in an object-centric state-space is valuable for obtaining good predictive models (Watters et al., 2017). Existing works in this area, such as SQAIR (Kosiorek et al., 2018), DDPAE (Hsieh et al., 2018), R-NEM (Van Steenkiste et al., 2018), and COBRA (Watters et al., 2019) have explored these concepts, but have not demonstrated realistic long term video predictions on par with supervised approaches to modeling physics.

To push the limits of unsupervised learning of physical interactions, we propose *STOVE*, a structured, object-aware video model. With STOVE, we combine image and physics modeling into a single state-space modelwhich explicitly reasons about object positions and velocities. It is trained end-to-end on pure video data in a self-supervised fashion and learns to detect objects, to model their interactions, and to predict future states and observations. To facilitate learning via variational inference in this model, we provide a novel inference architecture, which reuses the learned generative physics model in the variational distribution. As we will demonstrate, our model generates convincing rollouts over hundreds of time steps, outperforms other video modeling approaches, and approaches the performance of the supervised baseline which has access to the ground truth object states.

Moving beyond unsupervised learning, we also demonstrate how STOVE can be employed for model-based reinforcement learning (RL). Model-based approaches to RL have long been viewed

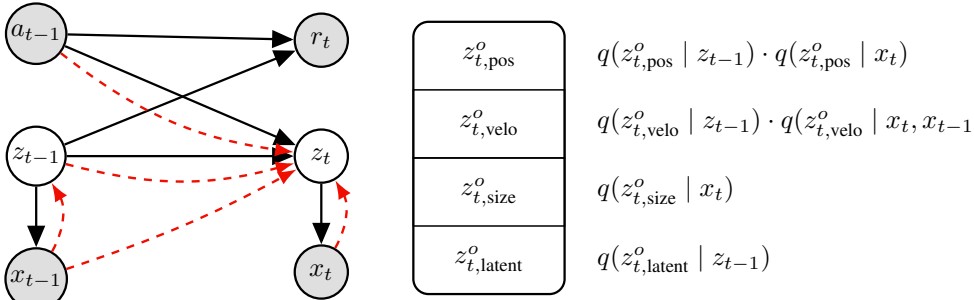

| | |
|---|---|
| $z_{t,\text{pos}}^o$ | $q(z_{t,\text{pos}}^o \mid z_{t-1}) \cdot q(z_{t,\text{pos}}^o \mid x_t)$ |
| $z_{t,\text{velo}}^o$ | $q(z_{t,\text{velo}}^o \mid z_{t-1}) \cdot q(z_{t,\text{velo}}^o \mid x_t, x_{t-1})$ |
| $z_{t,\text{size}}^o$ | $q(z_{t,\text{size}}^o \mid x_t)$ |
| $z_{t,\text{latent}}^o$ | $q(z_{t,\text{latent}}^o \mid z_{t-1})$ |

Figure 2: (Left) Depiction of the graphical model underlying STOVE. Black arrows denote the generative mechanism and red arrows the inference procedure. The variational distribution $q(z_t \mid z_{t-1}, x_t, x_{t-1})$ is formed by combining predictions from the dynamics model $p(z_t \mid z_{t-1})$ and the object detection network $q(z_t \mid x_t)$. For the RL domain, our approach is extended by action conditioning and reward prediction. (Right) Components of $z_t^o$ and corresponding variational distributions. Note that the velocities are estimated based on the change in positions between timesteps, inducing a dependency on $x_{t-1}$.

as a potential remedy to the often prohibitive sample complexity of model-free RL, but obtaining learned models of sufficient quality has proven difficult in practice (Sutton & Barto, 2011). By conditioning state predictions on actions and adding reward predictions to our dynamics predictor, we extend our model to the RL setting, allowing it to be used for search or planning. Our empirical evidence shows that an actor based on Monte-Carlo tree search (MCTS) (Coulom, 2007) on top of our model is competitive to model-free approaches such as Proximal Policy Optimization (PPO) (Schulman et al., 2017), while only requiring a fraction of the samples.

We proceed by introducing the two main components of STOVE: a structured image model and a dynamics model. We show how to perform joint inference and training, as well as how to extend the model to the RL setting. We then present our experimental evaluation, before touching on further related work and concluding.

## 2 STRUCTURED OBJECT-AWARE VIDEO MODELING

We approach the task of modeling a video with frames $x_1, \ldots, x_T$ from a probabilistic perspective, assuming a sequence of Markovian latent states $z_1, \ldots, z_T$, which decompose into the properties of a fixed number $O$ of objects, i.e. $z_t = (z_t^1, \ldots, z_t^O)$. In the spirit of compositionality, we propose to specify and train such a model by explicitly combining a dynamics prediction model $p(z_{t+1} \mid z_t)$ and a scene model $p(x_t \mid z_t)$. This yields a state-space model, which can be trained on pure video data, using variational inference and an approximate posterior distribution $q(z \mid x)$. Our model differs from previous work that also follows this methodology, most notably SQAIR and DDPAE, in three major ways:

- We propose a more compact architecture for the variational distribution $q(z \mid x)$, which reuses the dynamics model $p(z_{t+1} \mid z_t)$, and avoids the costly double recurrence across time and objects which was present in previous work.

- We parameterize the dynamics model using a graph neural network, taking advantage of the decomposed nature of the latent state $z$.

- Instead of treating each $z_t^o$ as an arbitrary latent code, we explicitly reserve the first six slots of this vector for the object's position, size, and velocity, each in $x, y$ direction, and use this information for the dynamics prediction task. We write $z_t^o = (z_{t,\text{pos}}^o, z_{t,\text{size}}^o, z_{t,\text{velo}}^o, z_{t,\text{latent}}^o)$.

We begin by briefly introducing the individual components before discussing how they are combined to form our state-space model. Fig. 1 visualises the computational flow of STOVE's inference and generative routines, Fig. 2 (left) specifies the underlying graphical model.

## 2.1 Object-based Modeling of Images using Sum-Product Attend-Infer-Repeat

A variety of object-centric image models have recently been proposed, many of which are derivatives of attend-infer-repeat (AIR) (Eslami et al., 2016). AIR postulates that each image consists of a set of objects, each of which occupies a rectangular region in the image, specified by positional parameters $z^o_{\text{where}} = (z^o_{\text{pos}}, z^o_{\text{size}})$. The visual content of each object is described by a latent code $z^o_{\text{what}}$. By decoding $z^o_{\text{what}}$ with a neural network and rendering the resulting image patches in the prescribed location, a generative model $p(x \mid z)$ is obtained. Inference is accomplished using a recurrent neural network, which outputs distributions over the latent objects $q(z^o \mid x)$, attending to one object at a time. AIR is also capable of handling varying numbers of objects, using an additional set of latent variables.

Sum-Product Attend-Infer-Repeat (SuPAIR) (Stelzner et al., 2019) utilizes sum-product networks (SPNs) instead of a decoder network to directly model the distribution over object appearances. The tractable inference capabilities of the SPNs used in SuPAIR allow for the exact and efficient computation of $p(x \mid z_{\text{where}})$, effectively integrating out the appearance parameters $z_{\text{what}}$ analytically. This has been shown to drastically accelerate learning, as the reduced inference workload significantly lowers the variance of the variational objective. Since the focus of SuPAIR on interpretable object parameters fits our goal of building a structured video model, we apply it as our image model $p(x_t \mid z_t)$. Similarly, we use a recurrent inference network as in SuPAIR to model $q(z_{t,\text{where}} \mid x_t)$. For details on SuPAIR, we refer to Stelzner et al. (2019).

## 2.2 Modeling Physical Interactions using Graph Neural Networks

In order to successfully capture complex dynamics, the state transition distribution $p(z_{t+1} \mid z_t) = p(z^1_{t+1}, \ldots, z^O_{t+1} \mid z^1_t, \ldots, z^O_t)$ needs to be parameterized using a flexible, non-linear estimator. A critical property that should be maintained in the process is *permutation invariance*, i.e., the output should not depend on the order in which objects appear in the vector $z_t$. This type of function is well captured by graph neural networks, cf. (Santoro et al., 2017), which posit that the output should depend on the sum of pairwise interactions between objects. Graph neural networks have been extensively used for modeling physical processes in supervised scenarios (Battaglia et al., 2016; 2018; Sanchez-Gonzalez et al., 2018; Zhou et al., 2018).

Following this line of work, we build a dynamics model of the basic form

$$\hat{z}^o_{t+1,\text{pos}}, \hat{z}^o_{t+1,\text{velo}}, \hat{z}^o_{t+1,\text{latent}} = f\left( g(z^o_t) + \sum_{o' \neq o} \alpha(z^o_t, z^{o'}_t) h(z^o_t, z^{o'}_t) \right) \tag{1}$$

where $f, g, h, \alpha$ represent functions parameterized by dense neural networks. $\alpha$ is an attention mechanism outputting a scalar which allows the network to focus on specific object pairs. We assume a constant prior over the object sizes, i.e., $\hat{z}^o_{t+1,\text{size}} = z^o_{t,\text{size}}$. The full state transition distribution is then given by the Gaussian $p(z^o_{t+1} \mid z^o_t) = \mathcal{N}(\hat{z}^o_{t+1}, \sigma)$, using a fixed $\sigma$.

## 2.3 Joint State-Space Model

Next, we assemble a state-space model from the two separate models for image modeling and physics prediction. The interface between the two components are the latent positions and velocities. The scene model infers them from images and the physics model propagates them forward in time. Combining the two yields the state-space model $p(x, z) = p(z_0)p(x_0 \mid z_0) \prod_t p(z_t \mid z_{t-1})p(x_t \mid z_t)$. To initialize the state, we model $p(z_0, z_1)$ using simple uniform and Gaussian distributions. Details are given in Appendix C.3.

Our model is trained on given video sequences $x$ by maximizing the evidence lower bound (ELBO) $\mathbb{E}_{q(z|x)}[\log p(x, z) - \log q(z \mid x)]$. This requires formulating a variational distribution $q(z \mid x)$ to approximate the true posterior $p(z \mid x)$. A natural approach is to factorize this distribution over time, i.e. $q(z \mid x) = q(z_0 \mid x_0) \prod_t q(z_t \mid z_{t-1}, x_t)$, resembling a Bayesian filter. The distribution $q(z_0 \mid x_0)$ is then readily available using the inference network provided by SuPAIR.

The formulation of $q(z_t \mid z_{t-1}, x_t)$, however, is an important design decision. Previous work, including SQAIR and DDPAE, have chosen to unroll this distribution over objects, introducing a

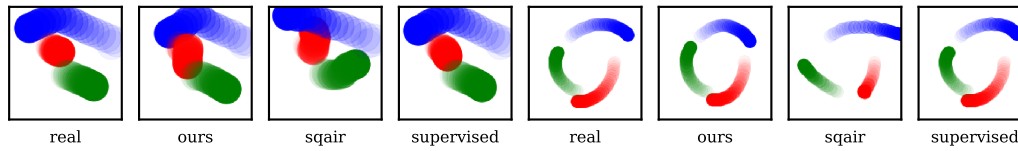

Figure 3: Visualisation of object positions from the real environment and predictions made by our model, SQAIR, and the supervised baseline, for the billiards and gravity environment after the first 8 frames were given. Our model achieves realistic behaviour, outperforms the unsupervised baselines, and approaches the quality of the supervised baseline, despite being fully unsupervised. For full effect, the reader is encouraged to watch animated versions of the sequences in repository github.com/jlko/STOVE. (Best viewed in color.)

costly double recurrence over time and objects, requiring $T \cdot O$ sequential recurrence steps in total. This increases the variance of the gradient estimate, slows down training, and hampers scalability. Inspired by Becker-Ehmck et al. (2019), we avoid this cost by *reusing* the dynamics model for the variational distribution. First, we construct the variational distribution $q(z_{t,\mathrm{pos}}^o \mid z_{t-1}^o)$ by slightly adjusting the dynamics prediction $p(z_{t,\mathrm{pos}}^o \mid z_{t-1}^o)$, using the same mean values but separately predicted standard deviations. Together with an estimate for the *same* object by the object detection network $q(z_{t,\mathrm{pos}}^o \mid x_t)$, we construct a joint estimate by multiplying the two Gaussians and renormalizing, yielding another Gaussian:

$$q(z_{t,\mathrm{pos}}^o \mid z_{t-1}, x_t) \propto q(z_{t,\mathrm{pos}}^o \mid z_{t-1}) \cdot q(z_{t,\mathrm{pos}}^o \mid x_t). \tag{2}$$

Intuitively, this results in a distribution which reconciles the two proposals. A double recurrence is avoided since $q(z_t \mid x_t)$ does not depend on previous timesteps and may thus be computed in parallel for all frames. Similarly, $q(z_t \mid z_{t-1})$ may be computed in parallel for all objects, leading to only $T + O$ sequential recurrence steps total. An additional benefit of this approach is that the information learned by the dynamics network is reused for inference — if $q(z_t \mid x_t, z_{t-1})$ were just another neural network, it would have to essentially relearn the environment's dynamics from scratch, resulting in a waste of parameters and training time. A further consequence is that the image likelihood $p(x_t \mid z_t)$ is backpropagated through the dynamics model, which has been shown to be beneficial for efficient training (Karl et al., 2017; Becker-Ehmck et al., 2019). The same procedure is applied to reconcile velocity estimates from the two networks, where for the image model, velocities $z_{t,\mathrm{velo}}^o$ are estimated from position differences between two consecutive timesteps. The object scales $z_{t,\mathrm{scale}}^o$ are inferred solely from the image model. The latent states $z_{t,\mathrm{latent}}^o$ increase the modelling capacity of the dynamics network, are initialised to zero-mean Gaussians, and do not interact with the image model. This then gives the inference procedure for the full latent state $z_t^o = (z_{t,\mathrm{pos}}^o, z_{t,\mathrm{size}}^o, z_{t,\mathrm{velo}}^o, z_{t,\mathrm{latent}}^o)$, as illustrated in Fig. 2 (right).

Despite its benefits, this technique has thus far only been used in environments with a single object or with known state information. A challenge when applying it in a multi-object video setting is to match up the proposals of the two networks. Since the object detection RNN outputs proposals for object locations in an indeterminate order, it is not immediately clear how to find the corresponding proposals from the dynamics network. We have, however, found that a simple matching procedure results in good performance: For each $z_t$, we assign the object order that results in the minimal difference of $||z_{t,\mathrm{pos}} - z_{t-1,\mathrm{pos}}||$, where $||\cdot||$ is the Euclidean norm. The resulting Euclidean bipartite matching problem can be solved in cubic time using the classic Hungarian algorithm (Kuhn, 1955).

### 2.4 Conditioning on Actions

In reinforcement learning, an agent interacts with the environment sequentially through actions $a_t$ to optimize a cumulative reward $r$. To extend STOVE to operate in this setting, we make two changes, yielding a distribution $p(z_t, r_t \mid z_{t-1}, a_{t-1})$.

First, we condition the dynamics model on actions $a_t$, enabling a conditional prediction based on both state and action. To keep the model invariant to the order of the input objects, the action information is concatenated to each object state $z_{t-1}^o$ before they are fed into the dynamics model. The model has to learn on its own which of the objects in the scene are influenced by the actions. To facilitate this, we have found it helpful to also concatenate appearance information from the

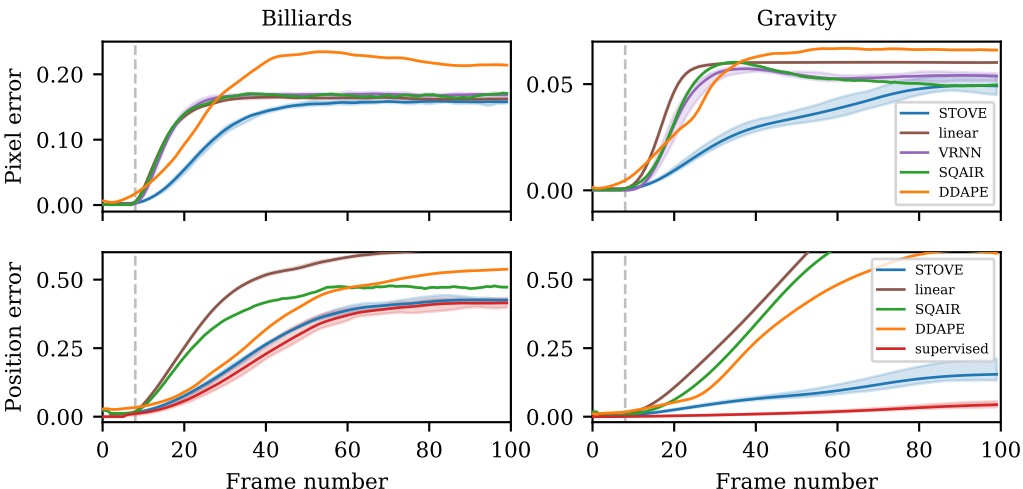

Figure 4: Mean test set performance of our model compared to baselines. Our approach (STOVE) clearly outperforms all unsupervised baselines and is almost indistinguishable from the supervised dynamics model on the billiards task. (Top) Mean squared errors over all pixels in the video prediction setting (the lower, the better). (Bottom) Mean Euclidean distances between predicted and true positions (the lower, the better). All position and pixel values are in $[0, 1]$. In all experiments, the first eight frames are given, all remaining frames are then conditionally generated. The shading indicates the max and min values over multiple training runs with identical hyperparameters. (Best viewed in color.)

extracted object patches to the object state. While this patch-wise code could, in general, be obtained using some neural feature extractor, we achieved satisfactory performance by simply using the mean values per color channel when given colored input.

The second change to the model is the addition of reward prediction. In many RL environments, rewards depend on the interactions between objects. Therefore, the dynamics prediction architecture, presented in Eq. 1, is well suited to also predict rewards. We choose to share the same encoding of object interactions between reward and dynamics prediction and simply apply two different output networks ($f$ in Eq. 1) to obtain the dynamics and reward predictions. The total model is again optimized using the ELBO, this time including the reward likelihood $p(r_t \mid z_{t-1}, a_{t-1})$.

## 3 EXPERIMENTAL EVIDENCE

In order to evaluate our model, we compare it to baselines in three different settings: First, pure video prediction, where the goal is to predict future frames of a video given previous ones. Second, the prediction of future object positions, which may be relevant for downstream tasks. Third, we extend one of the video datasets to a reinforcement learning task and investigate how our physics model may be utilized for sample-efficient, model-based reinforcement learning. With this paper, we also release a PyTorch implementation of STOVE.[1]

### 3.1 VIDEO AND STATE MODELING

Inspired by Watters et al. (2017), we consider grayscale videos of objects moving according to physical laws. In particular, we opt for the commonly used bouncing billiards balls dataset, as well as a dataset of gravitationally interacting balls. For further details on the datasets, see Appendix D. When trained using a single GTX 1080 Ti, STOVE converges after about 20 hours. As baselines, we compare to VRNNs (Chung et al., 2015), SQAIR (Kosiorek et al., 2018), and DDPAE (Hsieh et al., 2018). To allow for a fair comparison, we fix the number of objects predicted by SQAIR and DDPAE

---

[1] The code can be found in the GitHub repository github.com/jlko/STOVE. It also contains animated versions of the videos predicted by our model and the baselines.

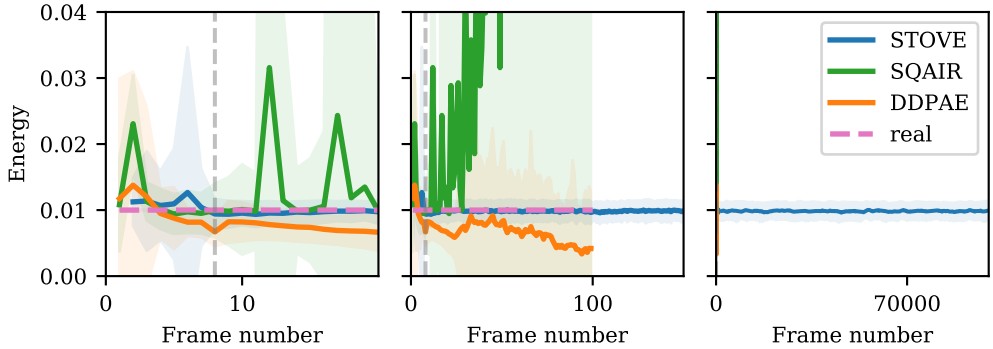

Figure 5: Comparison of the kinetic energies of the rollouts predicted by the models, computed based on position differences between successive states. Only STOVE's predictions reflect the conservation of total kinetic energy in the billiards data set. This is a quantitive measure of the convincing physical behavior in the rollout videos. (Left, center) Averages are over 300 trajectories from the test set. Shaded regions indicate one standard deviation. STOVE correctly predicts trajectories with constant energy, whereas SQAIR and DDPAE quickly diverge. (Right) Rolling average over a single, extremely long-term run. We conjecture that STOVE predicts physical behavior indefinitely. (Best viewed in color.)

to the correct amount. Furthermore, we compare to a supervised baseline: Here, we consider the ground truth positions and velocities to be fully observed, and train our dynamics model on them, resembling the setting of Battaglia et al. (2016). Since our model needs to infer object states from pixels, this baseline provides an upper bound on the predictive performance we can hope to achieve with our model. In turn, the size of the performance gap between the two is a good indicator of the quality of our state-space model. We also report the results obtained by combining our image model with a simple linear physics model, which linearly extrapolates the objects' trajectories. Since VRNN does not reason about object positions, we only evaluate it on the video prediction task. Similarly, the supervised baseline does not reason about images and is considered for the position prediction task only. For more information on the baselines, see Appendix E.

Fig. 4 depicts the reconstruction and prediction errors of the various models: Each model is given eight frames of video from the test set as input, which it then reconstructs. Conditioned on this input, the models predict the object positions or resulting video frames for the following 92 timesteps. The predictions are evaluated on ground truth data by computing the mean squared error between pixels and the Euclidean distance between positions based on the best available object matching. We outperform all baselines on both the state and the image prediction task by a large margin. Additionally, we perform strikingly close to the supervised model.

For the gravitational data, the prediction task appears easier, as all models achieve lower errors than on the billiards task. However, in this regime of easy prediction, precise access to the object states becomes more important, which is likely the reason why the gap between our approach and the supervised baseline is slightly more pronounced. Despite this, STOVE produces high-quality rollouts and outperforms the unsupervised baselines.

Table 1 underlines these results with concrete numbers. We also report results for three ablations of STOVE, which are obtained by (a) training a separate dynamics networks for inference with the same graph neural network architecture, instead of sharing weights with the generative model as argued for in section 2.3, (b) no longer explicitly modelling velocities $z_{velo}$ in the state, and (c) removing the latent state variables $z_{latent}$. The ablation study shows that each of these components contributes positively to the performance of STOVE. See Appendix F for a comparison of training curves for the ablations.

Fig. 3 illustrates predictions on future object positions made by the models, after each of them was given eight consecutive frames from the datasets. Visually, we find that STOVE predicts physically plausible sequences over long timeframes. This desirable property is not captured by the rollout error: Due to the chaotic nature of our environments, infinitesimally close initial states diverge quickly and a model which perfectly follows the ground truth states cannot exist. After this divergence has

Table 1: Predictive performance of our approach, the baselines, and ablations (lower is better, best unsupervised values are bold). STOVE outperforms all unsupervised baselines and is almost indistinguishable from the supervised model on the billiards task. The values are computed by summing the prediction errors presented in Fig. 4 in the time interval $t \in [9, 18]$, i.e., the first ten predicted timesteps. In parentheses, standard deviations across multiple training runs are given.

|  | Billiards (pixels) | Billiards (positions) | Gravity (pixels) | Gravity (positions) |
|---|---|---|---|---|
| STOVE (ours) | **0.240(14)** | **0.418(20)** | **0.040(3)** | **0.142(7)** |
| VRNN | 0.526(14) | – | 0.055(12) | – |
| SQAIR | 0.591 | 0.804 | 0.070 | 0.194 |
| DDPAE | 0.405 | 0.482 | 0.120 | 0.298 |
| Linear | 0.844(5) | 1.348(15) | 0.196(2) | 0.493(4) |
| Supervised | – | 0.232(37) | – | 0.013(2) |
| Abl: Double Dynamics | 0.262 | 0.458 | 0.042 | 0.154 |
| Abl: No Velocity | 0.272 | 0.460 | 0.053 | 0.174 |
| Abl: No Latent | 0.338 | 0.050 | 0.089 | 0.235 |

occurred, the rollout error no longer provides any information on the quality of the learned physical behavior. We therefore turn to investigating the total kinetic energy of the predicted billiards trajectories. Since the collisions in the training set are fully elastic and frictional forces are not present, the initial energy should be conserved. Fig. 5 shows the kinetic energies of trajectories predicted by STOVE and its baselines, computed based on the position differences between consecutive timesteps. While the energies of SQAIR and DDPAE diverge quickly in less than 100 frames, the mean energies of STOVE's rollouts stay constant and are good estimates of the true energy. We have confirmed that STOVE predicts constant energies – and therefore displays realistic looking behavior – for at least 100 000 steps. This is in stark contrast to the baselines, which predict teleporting, stopping, or overlapping objects after less than 100 frames. In the billiards dataset used by us and the literature, the total energy is the same for all sequences in the training set. See Appendix B for a discussion of how STOVE handles diverse energies.

## 3.2 MODEL-BASED CONTROL

To explore the usefulness of STOVE for reinforcement learning, we extend the billiards dataset into a reinforcement learning task. Now, the agent controls one of the balls using nine actions, which correspond to moving in one of the eight (inter)cardinal directions and staying at rest. The goal is to avoid collisions with the other balls, which elastically bounce off of each other, the walls, and the controlled ball. A negative reward of $-1$ is given whenever the controlled ball collides with one of the others. To allow the models to recognize the object controlled by the agents we now provide it with RGB input in which the balls are colored differently. Starting with a random policy, we iteratively gather observations from the environment, i.e. sequences of images, actions, and rewards. Using these, we train our model as described in Sec. 2.4. To obtain a policy based on our world model, we use Monte-Carlo tree search (MCTS), leveraging our model as a simulator for planning. Using this policy, we gather more observations and apply them to refine the world model. As an upper bound on the performance achievable in this manner, we report the results obtained by MCTS when the real environment is used for planning. As a model-free baseline, we consider PPO (Schulman et al., 2017), which is a state-of-the-art algorithm on comparable domains such as Atari games. To explore the effect of the availability of state information, we also run PPO on a version of the environment in which, instead of images, the ground-truth object positions and velocities are observed directly.

Learning curves for each of the agents are given in Fig. 6 (left), reported at intervals of 10 000 samples taken from the environment, up to a total of 130 000. For our model, we collect the first 50 000 samples using a random policy to provide an initial training set. After that, the described training loop is used, iterating between collecting 10 000 observations using an MCTS-based policy and refining the model using examples sampled from the pool of previously seen observations. After 130 000 samples, PPO has not yet seen enough samples to converge, whereas our model quickly learns to meaningfully model the environment and thus produces a better policy at this stage. Even when PPO is trained on ground truth states, MCTS based on STOVE remains comparable.

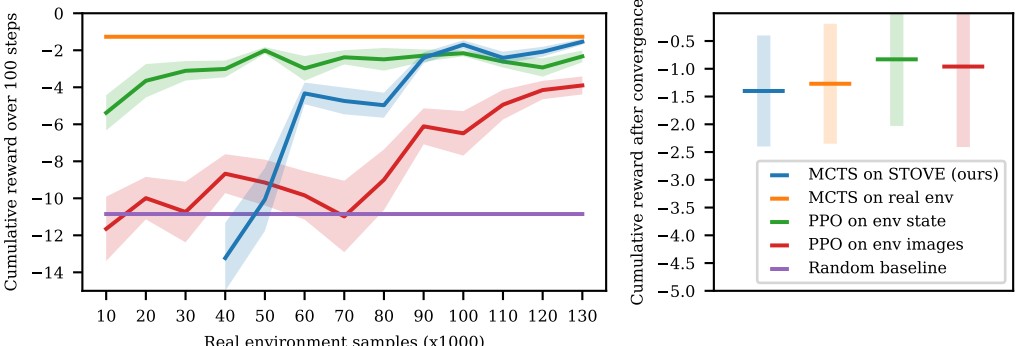

Figure 6: Comparison of all models on sample efficiency and final performance. (Left) Mean cumulative reward over 100 steps on the environment, averaged over 100 environments, using the specified policy. The shaded regions correspond to one-tenth of a standard deviation. In addition to the training curves, two constant baselines are shown, one representing a random policy and one corresponding to the MCTS based policy when using the real environment as a simulator. (Right) Final performance of all approaches, after training each model to convergence. The shaded region corresponds to one standard deviation. (Best viewed in color.)

After training each model to convergence, the final performance of all approaches is reported in Fig. 6 (right). In this case, PPO achieves slightly better results, however it only converges after training for approximately $4\,000\,000$ steps, while our approach only uses $130\,000$ samples. After around $1\,500\,000$ steps, PPO does eventually surpass the performance of STOVE-based MCTS. Additionally, we find that MCTS on STOVE yields almost the same performance as on the real environment, indicating that it can be used to anticipate and avoid collisions accurately.

## 4 RELATED WORK

Multiple lines of work with the goal of video modeling or prediction have emerged recently. Prominently, the supervised modeling of physical interactions from videos has been investigated by Fragkiadaki et al. (2015), who train a model to play billiards with a single ball. Similarly, graph neural networks have been trained in a supervised fashion to predict the dynamics of objects from images (Watters et al., 2017; Sanchez-Gonzalez et al., 2018; Sun et al., 2018; 2019) or ground truth states (Kipf et al., 2018; Wang et al., 2018; Chang et al., 2017). A number of works learn object interactions in games in terms of rules instead of continuous dynamics (Guzdial et al., 2017; Ersen & Sariel, 2014). Janner et al. (2019) show successful planning based on learned interactions, but assume access to image segmentations. Several unsupervised approaches address the problem by fitting the parameters of a physics engine to data (Jaques et al., 2019; Wu et al., 2016; 2015). This necessitates specifying in advance which physical laws govern the observed interactions. In the fully unsupervised setting, mainly unstructured variational approaches have been explored (Babaeizadeh et al., 2017; Chung et al., 2015; Krishnan et al., 2015). However, without the explicit notion of objects, their performance in scenarios with interacting objects remains limited. Nevertheless, unstructured video models have recently been applied to model-based RL and have been shown to improve sample efficiency when used as a simulator for the real environment (Oh et al., 2015; Kaiser et al., 2020).

Only a small number of works incorporate objects into unsupervised video models. Xu et al. (2019) and Ehrhardt et al. (2018) take non-probabilistic autoencoding approaches to discovering objects in real-world videos. COBRA (Watters et al., 2019) represents a model-based RL approach based on MONet, but is restricted to environments with non-interacting objects and only uses one-step search to build its policy. Closest to STOVE are a small number of probabilistic models, namely SQAIR (Kosiorek et al., 2018), R-NEM (Van Steenkiste et al., 2018; Greff et al., 2017), and DDPAE (Hsieh et al., 2018). R-NEM learns a mixture model via expectation-maximization unrolled through time and handles interactions between objects in a factorized fashion. However, it lacks an explicitly

structured latent space, and requires noise in the input data to avoid local minima. Both DDPAE and SQAIR extend the AIR approach to work on videos using standard recurrent architectures. As discussed, this introduces a double recurrence over objects and time, which is detrimental for performance. However, SQAIR is capable of handling a varying number of objects, which is not something we consider in this paper.

## 5 CONCLUSION

We introduced STOVE, a structured, object-aware model for unsupervised video modeling and planning. It combines recent advances in unsupervised image modeling and physics prediction into a single compositional state-space model. The resulting joint model explicitly reasons about object positions and velocities, and is capable of generating highly accurate video predictions in domains featuring complicated non-linear interactions between objects. As our experimental evaluation shows, it outperforms previous unsupervised approaches and even approaches the performance and visual quality of a supervised model.

Additionally, we presented an extension of the video learning framework to the RL setting. Our experiments demonstrate that our model may be utilized for sample-efficient model-based control in a visual domain, making headway towards a long standing goal of the model-based RL community. In particular, STOVE yields good performance with more than one order of magnitude fewer samples compared to the model-free baseline, even when paired with a relatively simple planning algorithm like MCTS.

At the same time, STOVE also makes several assumptions for the sake of simplicity. Relaxing them provides interesting avenues for future research. First, we assume a fixed number of objects, which may be avoided by performing dynamic object propagation and discovery like in SQAIR. Second, we have inherited the assumption of rectangular object masks from AIR. Applying a more flexible model such as MONet (Burgess et al., 2019) or GENESIS (Engelcke et al., 2020) may alleviate this, but also poses additional challenges, especially regarding the explicit modeling of movement. Finally, the availability of high-quality learned state-space models enables the use of more sophisticated planning algorithms in visual domains (Chua et al., 2018). In particular, by combining planning with policy and value networks, model-free and model-based RL may be integrated into a comprehensive system (Buckman et al., 2018).

**Acknowledgments.**   The authors thank Adam Kosiorek for his assistance with the SQAIR experiments and Emilien Dupont for helpful discussions about conservation laws in dynamics models. KK acknowledges the support of the Rhine-Main universities' network for "Deep Continuous-Discrete Machine Learning" (DeCoDeML).

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

## A    RECONSTRUCTIONS: SPRITES DATA

SuPAIR does not need a latent description of the objects' appearances. Nevertheless, object reconstructions can be obtained by using a variant of approximate MPE (most probable explanation) in the sum-product networks as proposed by Vergari et al. (2018). We follow the AIR approach and reconstruct each object separately and paste it into the canvas using spatial transformers. Unlike AIR, SuPAIR explicitly models the background using a separate background SPN. A reconstruction of the background is also obtained using MPE.

To demonstrate the capabilities of our image model, we also trained our model on a variant of the gravity data in which the round balls were replaced by a random selection of four different sprites of the same size. Fig. 7 shows the reconstructions obtained from SuPAIR when trained on these more complex object shapes.

## B    STUDY OF ENERGIES

As discussed in Sec. 3.1, the energies of the ground truth data were constant for all sequences during the training of STOVE. However, initial velocities are drawn from a random normal distribution. This is the standard procedure of generating the bouncing balls data set as used by previous publications. Under these circumstances, STOVE does indeed learn to discover and replicate the total energies of the system, while SQAIR and DDPAE do not. Even if trained on constant energy data, STOVE does to some extent generalise to unseen energies. Observed velocities and therefore total energies are highly correlated with the true total kinetic energies of the sequences. However as prediction starts, STOVE quickly regresses to the energy of the training set, see Fig. 8 (left). If trained on a dataset of diverse total energies, the performance of modelling sequences of different energies increases, see Fig. 8 (right). Rollouts now initially represent the true energy of the observed sequence, although this estimate of the true energy diverges over a time span of around 500 frames to a constant but wrong energy value. This is an improvement over the model trained on constant energy data, where the regression to the training data energy happens much quicker within around 10 frames. Note that this does not drastically decrease the visual quality of the rollouts as the change of total energy over 500 frames is gradual enough. We leave the reliable prediction of rollouts with physically valid constant energy for sequences of varying energies for future work.

## C    MODEL DETAILS

Here, we present additional details on the architecture and hyperparameters of STOVE.

### C.1    INFERENCE ARCHITECTURE

The object detection network for $q(z_{t,\text{where}} \mid x_t)$ is realised by an LSTM (Hochreiter & Schmidhuber, 1997) with 256 hidden units, which outputs the mean and standard deviation of the objects' two-dimensional position and size distributions, i.e. $q(z^o_{t,\text{pos, size}} \mid x_t)$ with $2 \cdot 2 \cdot 2 = 8$ parameters per object. Given such position distributions for two consecutive timesteps $q(z_{t-1,\text{pos}} \mid x_{t-1}), q(z_{t,\text{pos}} \mid x_t)$, with parameters $\mu_{z^o_{t-1,\text{pos}}}, \sigma_{z^o_{t-1,\text{pos}}}, \mu_{z^o_{t,\text{pos}}}, \sigma_{z^o_{t,\text{pos}}}$, the following velocity estimate based on the difference in position is constructed:

$$q(z^o_{t,\text{velo}} \mid x_t, x_{t-1}) = \mathcal{N}(\mu_{z^o_{t,\text{pos}}} - \mu_{z^o_{t-1,\text{pos}}}, \sigma^2_{z^o_{t,\text{pos}}} + \sigma^2_{z^o_{t-1,\text{pos}}}).$$

As described in Sec. 2.3, positions and velocities are also inferred from the dynamics model as $q(z^o_{t,\text{pos}} \mid z_{t-1})$ and $q(z^o_{t,\text{velo}} \mid z_{t-1})$. A joint estimate, including information from both image model and dynamics prediction, is obtained by multiplying the respective distributions and renormalizing. Since both $q$-distributions are Gaussian, the normalized product is again Gaussian, with mean and

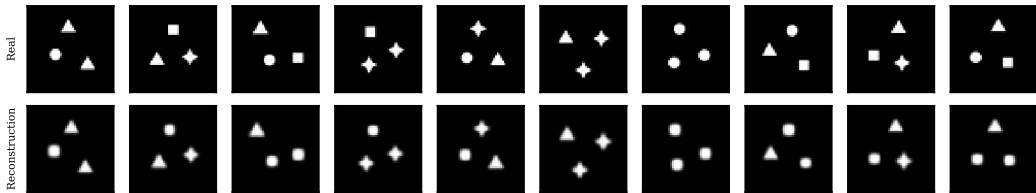

Figure 7: Reconstructions obtained from our image model when using more varied shapes.

standard deviation are given by

$$
\begin{aligned}
q(z_t \mid x_t, z_{t-1}) &\propto q(z_t \mid x_t) \cdot q(z_t \mid z_{t-1}) \\
&= \mathcal{N}(z_t; \mu_{t,i}, \sigma_{t,i}^2) \cdot \mathcal{N}(z_t; \mu_{t,d}, \sigma_{t,d}^2) \\
&= \mathcal{N}(z_t; \mu_t, \sigma_t^2) \\
\mu_t &= \frac{\sigma_{t,d}^2 \mu_{t,i} + \sigma_{t,i}^2 \mu_{t,d}}{\sigma_{t,d}^2 + \sigma_{t,i}^2} \\
\frac{1}{\sigma_t^2} &= \frac{1}{\sigma_{t,d}^2} + \frac{1}{\sigma_{t,i}^2} \, ,
\end{aligned}
$$

where we relax our notation for readability $z_t \in [z_{t,\text{pos}}^o, z_{t,\text{velo}}^o]$ and the indices $i$ and $d$ refer to the parameters obtained from the image and dynamics model. This procedure is applied independently for the positions and velocities of each object.

For $z_{t,\text{latent}}^o$, we choose dimension 12, such that a full state $z_t^o = (z_{t,\text{pos}}^o, z_{t,\text{size}}^o, z_{t,\text{velo}}^o, z_{t,\text{latent}}^o)$ is 18-dimensional.

## C.2 GRAPH NEURAL NETWORK

The dynamics prediction is given by the following series of transformations applied to each input state of shape (`batch size, number of objects,` $l$), where $l = 16$, since currently, size information is not propagated through the dynamics prediction.

- $S_1$: Encode input state with linear layer $[l, 2l]$.
- $S_2$: Apply linear layer $[2l, 2l]$ to $S_1$ followed by ReLU non-linearity.
- $S_3$: Apply linear layer $[2l, 2l]$ to $S_2$ and add result to $S_2$. This gives the dynamics prediction without relational effects, corresponding to $g(z_t^o)$ in Eq. 1.
- $C_1$: The following steps obtain the relational aspects of dynamics prediction, corresponding to $h(z_t^o, z_t^{o'})$ in Eq. 1. Concatenate the encoded state $S_1^o$ pairwise with all state encoding, yielding a tensor of shape (`batch size, number of objects, number of objects,` $4l$).
- $C_2$: Apply linear layer $[4l, 4l]$ to $C_1$ followed by ReLU.
- $C_3$: Apply linear layer $[4l, 2l]$ to $C_2$ followed by ReLU.
- $C_4$: Apply linear layer $[2l, 2l]$ to $C_3$ and add to $C_3$.
- $A_1$: To obtain attention coefficients $\alpha(z_t^o, z_t^{o'})$, apply linear layer $[4l, 4l]$ to $C_1$ followed by ReLU.
- $A_2$: Apply linear layer $[4l, 2l]$ to $A_1$ followed by ReLU.
- $A_3$: Apply linear layer $[2l, 1]$ to $A_2$ and apply exponential function.
- $R_1$: Multiply $C_4$ with $A_3$, where diagonal elements of $A_3$ are masked out to ensure that $R_1$ only covers cases where $o \neq o'$.
- $R_2$: Sum over $R_1$ for all $o'$, to obtain tensor of shape (`batch size, number of objects,` $2l$). This is the relational dynamics prediction.

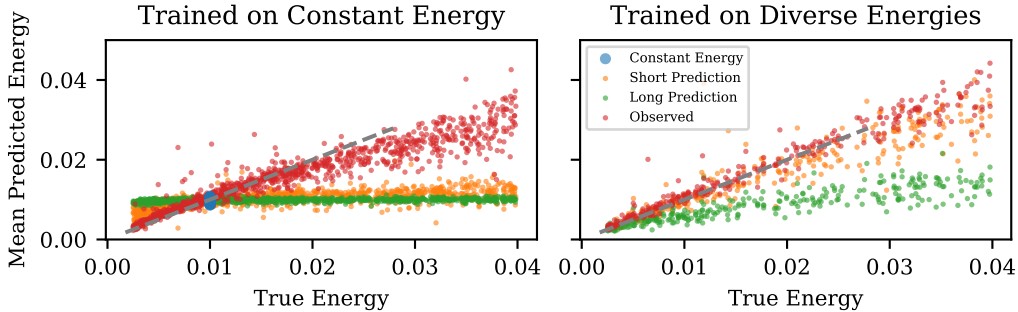

Figure 8: Mean kinetic energy observed/predicted by STOVE over true energy of the sequences. (left) STOVE is trained on sequences of constant kinetic energy. As can be seen from the blue scatter points, STOVE manages to predict sequences of arbitrary lengths which, on average, preserve the constant energy of the test set. When STOVE is applied to sequences of different energies, it manages to infer these energies from observed frames fairly well, with inaccuracies compounding at larger energies (red). In the following prediction, however, the mean predicted energies diverge quickly to the energy value of the training set (orange and green). (right) STOVE is now trained on sequences of varying energies. Compared to the constant energy training, energies from observed as well as predicted energies improve drastically. The predictions no longer immediately regress towards a specific value (orange). However after 100 frames, the quality of the predicted energies still regresses to a wrong value (green). (all) The observed values refers to energies obtained as the mean energy value over the six initially observed frames. The short (long) time frame refers to an energy obtained as the mean energy over the first 10 (100) frames of prediction. (Best viewed in color.)

- $D_1$: Sum relational dynamics $R_2$ and self-dynamics $S_3$, obtaining the input to $f$ in Eq. 1.

- $D_2$: Apply linear layer $[2l, 2l]$ to $D_1$ followed by tanh non-linearity.

- $D_3$: Apply linear layer $[2l, 2l]$ to $D_2$ followed by tanh non-linearity and add result to $D_2$.

- $D_4$: Concatenate $D_3$ and $S_1$, and apply linear layer $[4l, 2l]$ followed by tanh.

- $D_5$: Apply linear layer $[2l, 2l]$ to $D_4$ and add result to $D_4$ to obtain final dynamics prediction.

The output $D_5$ has shape (`batch size`, `number of objects`, $2l$), twice the size of means and standard deviations over the next predicted state.

For the model-based control scenario, the one-hot encoded actions (`batch size, action space`) are transformed with a linear layer [`action space, number of objects` · `encoding size`] and reshaped to (`action space, number of objects, encoding size`). The action embedding and the object appearances (`batch size, number of objects`, 3) are then concatenated to the input state. The rest of the dynamics prediction follows as above. The reward prediction consists of the following steps:

- $H_1$: Apply linear layer $[2l, 2l]$ to $D_1$ followed by ReLU.

- $H_2$: Apply linear layer $[2l, 2l]$ to $H_1$.

- $H_3$: Sum over object dimension to obtain tensor of shape (`batch size`, $l$).

- $H_4$: Apply linear layer $[l, l/2]$ to $H_3$ followed by ReLU.

- $H_5$: Apply linear layer $[l/2, l/4]$ to $H_4$ followed by ReLU.

- $H_5$: Apply linear layer $[l/4, l]$ to $H_4$ followed by a sigmoid non-linearity.

$H_5$ then gives the final reward prediction.

### C.3 STATE INITIALIZATION

In the first two timesteps, we cannot yet apply STOVE's main inference step $q(z_t \mid z_{t-1}, x_t, x_{t-1})$ as described above. In order to initialize the latent state over the first two frames, we apply a simplified architecture and only use a partial state at $t = 0$.

At $t = 0$, $z_0 \sim q(z_{0,(\text{pos, size})} \mid x_0)$ is given purely by the object detection network, since no previous states, which could be propagated, exist. $z_0$ is incomplete insofar as it does not contain velocity information or latents. At $t = 1$, $q(z_{1,\text{pos, size}} \mid x_1, x_0)$ is still given purely based on the object detection network. Note that for a dynamics prediction of $z_1$, velocity information at $t = 0$ would need to be available. However, at $t = 1$, velocities can be constructed based on the differences between the previously inferred object positions. We sample $z_{1,\text{latent}}$ from the prior Gaussian distribution to assemble the first full initial state $z_1$. At $t \geq 2$, the full inference network can be run: States are inferred both from the object detection network $q(z_t \mid x_t, x_{t-1})$ as well as propagated using the dynamics model $q(z_t \mid z_{t-1})$.

In the generative model, similar adjustments are made: $p(z_{0,\text{pos, size}})$ is given by a uniform prior, velocities and latents are omitted. At $t = 1$, velocities are sampled from a uniform distribution in planar coordinates $p(z_{1,\text{velo}})$ and positions are given by a simple linear dynamics model $p(z_{1,\text{pos}} \mid z_{0,\text{pos}}, z_{1,\text{velo}}) = \mathcal{N}(z_{0,\text{pos}} + z_{1,\text{velo}}, \sigma)$. Latents $z_{1,\text{latent}}$ are sampled from a Gaussian prior. Starting at $t = 2$, the full dynamics model is used.

### C.4 TRAINING PROCEDURE

Our model was trained using the Adam optimizer (Kingma & Ba, 2015), with a learning rate of $2 \times 10^{-3} \exp(-40 \times 10^{-3} \cdot \text{step})$ for a total of $83\,000$ steps with a batch size of 256.

## D DATA DETAILS

For the billiards and gravitational data, 1000 sequences of length 100 were generated for training. From these, subsequences of lengths 8 were sampled and used to optimize the ELBO. A test dataset of 300 sequences of length 100 was also generated and used for all evaluations. The pixel resolution of the dataset was $32 \times 32$ for the billiards data and $50 \times 50$ for the gravity data. All models for video prediction were learned on grayscale data, with objects of identical appearance. The $O = 3$ balls were initialised with uniformly random positions and velocities, rejecting configurations with overlap. They are rendered using anti-aliasing. The billiards data models the balls as circular objects, which perform elastic collision with each other or the walls of the environment. For the gravity data, the balls are modeled as point masses, where, following Watters et al. (2017), we clip the gravitational force to avoid slingshot effects. Also, we add an additional basin of attraction towards the center of the canvas and model the balls in their center off mass system to avoid drift. Velocities here are initialised orthogonal to the center of the canvas for a stabilising effect. For full details we refer to the file `envs.py` in the provided code.

## E BASELINES FOR VIDEO MODELING

Following Kosiorek et al. (2018), we experimented with different hyperparameter configurations for VRNNs. We varied the sizes of the hidden and latent states $[h, z]$, experimenting with the values $[256, 16]$, $[512, 32]$, $[1024, 64]$, and $[2048, 32]$. We found that increasing the model capacity beyond $[512, 32]$ did not yield large increases in performance, which is why we chose the configuration $[512, 32]$ for our experiments. Our VRNN implementation is written in PyTorch and based on `https://github.com/emited/VariationalRecurrentNeuralNetwork`.

SQAIR can handle a variable number of objects in each sequence. However, to allow for a fairer comparison to STOVE, we fixed the number of objects to the correct number. This means that in the first timestep, exactly three objects are discovered, which are then propagated in all following timesteps, without further discoveries. Our implementation is based on the original implementation provided by the authors at `https://github.com/akosiorek/sqair`.

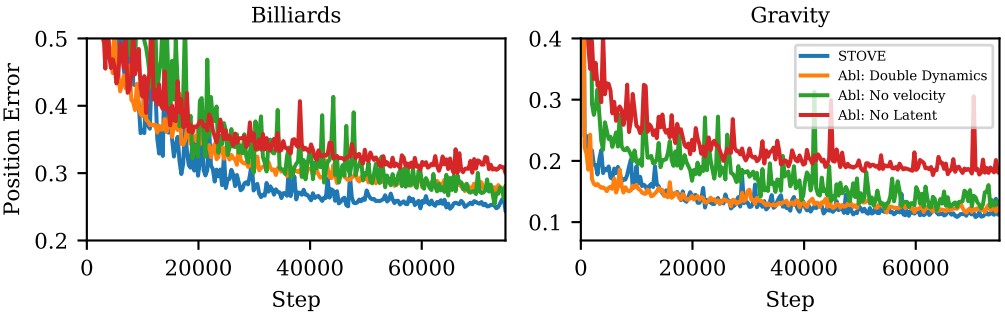

Figure 9: Displayed is the mean predicted position error over a rollout length of 8 frames as training progresses for the billiards (left) and gravity (right) scenario for STOVE and its ablations. (Best viewed in color.)

The DDPAE experiments were performed using the implementation available at `https://github.com/jthsieh/DDPAE-video-prediction`. Default parameters for training DDPAE with billiards datasets are provided with the code. However, the resolution of our billiards (32 pixels) and gravity (64 pixels) datasets is different to the resolution DDPAE expects (64 pixels). While we experimented with adjusting DDPAE parameters such as the latent space dimension to fit our different resolution, best results were obtained when bilinearly scaling our data to the resolution DDPAE expects. DDPAE was trained for $400\,000$ steps, which sufficed for convergence of the models' test set error.

The linear baseline was obtained as follows: For the first 8 frames, we infer the full model state using STOVE. We then take the last inferred positions and velocities of each object and predict future positions by assuming constant, uniform motions for each object. We do not allow objects to leave the frame, i. e. when objects reach the canvas boundary after some timesteps, they stick to it.

Since our dynamics model requires only object positions and velocities as input, it is trivial to construct a supervised baseline for our physics prediction by replacing the SuPAIR-inferred states with real, ground-truth states. On these, the model can then be trained in supervised fashion.

## F    TRAINING CURVES OF ABLATIONS

In Fig. 9 we display learning curves for STOVE and presented ablations. As mentioned in the main text, the ablations demonstrate the value of the reuse of the dynamics model, the explicit inclusion of a velocity value, and the presence of unstructured latent space in the dynamics model. (Best viewed in color.)

## G    DETAILS ON THE REINFORCEMENT LEARNING MODELS

Our MCTS implementation uses the standard UCT formulation for exploration/exploitation. The $c$ parameter is set to $1.$ in all our experiments. Since the environment does not provide a natural endpoint, we cut off all rollouts at a depth of 20 timesteps. We found this to be a good trade-off between runtime and accuracy.

When expanding a node on the true environment, we compute the result of the most promising action, and then start a rollout using a random policy from the resulting state. For the final evaluation, a total of 200 nodes are expanded. To better utilize the GPU, a slightly different approach is used for STOVE. When we expand a node in this setting, we predict the results of all actions simultaneously, and compute a rollout from each resulting position. In turn, only 50 nodes are expanded. To estimate the node value function, the average reward over all rollouts is propagated back to the root and each node's visit counter is increased by 1. Furthermore, we discount the reward predicted STOVE with a factor of $0.95$ per timestep to account for the higher uncertainty of longer rollouts. This is not done in the baseline running on the real environment, since it behaves deterministically.

For PPO, we employ a standard convolutional neural network as an actor-critic for the evaluation on images and a MLP for the evaluation on states. The image network consists of two convolutional layers, each using 32 output filters with a kernel size of 4 and 3 respectively and a stride of 2. The MLP consists of two fully connected layers with 128 and 64 hidden units. In both cases, an additional fully connected layer links the outputs of the respective base to an actor and a critic head. For the convolutional base, this linking layer employs 512 hidden units, for the MLP 64. All previously mentioned layers use rectified linear activations. The actor head predicts a probability distribution over next actions using a softmax activation function while the critic head outputs a value estimation for the current state using a linear prediction. We tested several hyperparameter configurations but found the following to be the most efficient one. To update the actor-critic architecture, we sample 32 trajectories of length 16 from separate environments in every batch. The training uses an Adam optimizer with a learning rate of $2 \times 10^{-4}$ and and $\epsilon$ value of $1 \times 10^{-5}$. The clipping parameter of PPO is set to $1 \times 10^{-1}$. We update the network for 4 epochs in each batch using 32 mini-batches of the sampled data. The value loss is weighted at $5 \times 10^{-1}$ and the entropy coefficient is set to $1 \times 10^{-2}$.

