# OpenReview forum: "Structured Object-Aware Physics Prediction for Video Modeling and Planning"
_ICLR.cc/2020/Conference — Accept (Poster)_

### Official Review · AnonReviewer1 · 2019-10-06
**Official Blind Review #1**

**Rating:** 6

**Review:**

This paper introduces a structured deep generative model for video frame prediction, with an object recognition model based on the Attend, Infer, Repeat (AIR) model by Eslami et al. (2016) and a graph neural network as a latent dynamics model. The model is evaluated on two synthetic physics simulation datasets (N-body gravitational systems and bouncing billiard balls) for next frame prediction and on a control task in the billiard domain. The model can produce accurate predictions for several time steps into the future and beats a variational RNN and SQAIR (sequential AIR variant) baseline, and is more sample-efficient than a model-free PPO agent in the control task.

Overall, the paper is well-structured, nicely written and addresses an interesting and challenging problem. The experiments use simple domains/problems, but give good insights into how the model performs.

Related work is covered to a satisfactory degree, but a discussion of some of the following closely related papers could improve the paper:
* Chang et al., A Compositional Object-Based Approach To Learning Physical Dynamics, ICLR 2017
* Greff et al., Neural Expectation Maximization, NeurIPS 2017
* Kipf et al., Neural Relational Inference for Interacting Systems, ICML 2018
* Greff et al., Multi-object representation learning with iterative variational inference, ICML 2019
* Sun et al., Actor-centric relation network, ECCV 2018
* Sun et al., Relational Action Forecasting, CVPR 2019
* Wang et al., NerveNet: Learning structured policy with graph neural networks, ICLR 2018
* Xu et al., Unsupervised discovery of parts, structure and dynamics, ICLR 2019
* Erhardt et al., Unsupervised intuitive physics from visual observations, ACCV 2018

In terms of clarity, the paper could be improved by making the used model architecture more explicit, e.g., by adding a model figure, and by providing an introduction to the SuPAIR model (Stelzner et al., 2019) — the authors assume that the reader is more or less familiar with this particular model. It is further unclear how exactly the input data is provided to the model; Figure 2 makes it seem that inputs are colored frames, section 3.1 mentions that inputs are grayscale videos (do all objects have the same appearance or different shades of gray?), which is in conflict with the statement on page 5 that the model is provided with mean values of input color channels. Please clarify.

In terms of novelty, the proposed modification of SQAIR (separating object detection and latent dynamics prediction) is novel and likely leads to a speed-up in training and evaluation. Using a Graph Neural Network for modeling latent physics is reasonable and has been shown to work on related problems before (see referenced work above and related work mentioned in the paper). Similarly, using such a model for planning/control is interesting and adds to the value of the paper, but has in related settings been explored before (e.g. Wang et al. (ICLR 2018) and Sanchez-Gonzalez (ICML 2018)).

Experimentally, it would be good to provide ablation studies (e.g. a different object detection module like AIR instead of SuPAIR, not splitting the latent variables into position, velocity, size etc.) and run-time comparisons (wall-clock time), as one of the main contributions of the paper is that the proposed model is claimed to be faster than SQAIR. The overall model predictions are (to my surprise) somewhat inaccurate, when looking at e.g. the billiard ball example in Figure 2. In Steenkiste et al. (ICLR 2018), roll-outs appear to be more accurate. Maybe a quantitative experimental comparison could help?

Why does the proposed model perform worse than a model-free PPO baseline when trained to convergence on the control task? What is missing to close this gap?

Do all objects have the same appearance (color/greyscale values) or are they unique in appearance? In the second case, a simpler encoder architecture could be used such as in Jaques et al. (2019) or Xu et al. (ICLR 2019).

Overall, I think that this paper addresses an important issue and is potentially of high interest to the community. Nonetheless I think that this paper needs a bit more work and at this point I recommend a weak reject.

Other comments:
* This sentence is unclear to me: “An additional benefit of this approach is that the information learned by the dynamics model is reused for inference — […]”
* What are the failure modes of the model? Where does it break down?
* How does the model deal with partial occlusion?

---------------------
UPDATE (after reading the author response and the revised manuscript): My questions and comments are addressed and the additional ablation studies and experimental results on energy conservation are convincing and insightful. I think the revised version of the paper meets the bar for acceptance at ICLR.


**Experience Assessment:**

I have published one or two papers in this area.

**Review Assessment: Checking Correctness Of Derivations And Theory:**

I did not assess the derivations or theory.

**Review Assessment: Checking Correctness Of Experiments:**

I assessed the sensibility of the experiments.

**Review Assessment: Thoroughness In Paper Reading:**

I read the paper at least twice and used my best judgement in assessing the paper.

---

> ### Author Response · Authors · 2019-11-08
> **Response to Reviewer 1**
>
> Dear Reviewer 1,
>
> thank you for your valuable feedback.
> Below, we give a detailed response to your questions and comments.
> Please also see the changes to the manuscript outlined in our top level comment.
>
> [Added a "Model Figure"]
> We have revised Figure 1 to include a visualisation of the latent space and the corresponding recognition distributions. We hope this clarifies the model structure.
>
> [Introduction to SuPAIR]
> We chose to omit details on SuPAIR as they are not required for understanding STOVE - in principle, any image model delivering a likelihood p(x | z_where) based on location information z_where could be used in its stead, including AIR. As said, we mainly chose SuPAIR due to its fast training times. If you have specific suggestions for what should be clarified about SuPAIR, we will be glad to do so.
>
> [Color vs. Grayscale]
> For the video modeling task, we use grayscale images in which all objects are the same shade of white. Color has been added to Figure 2 to make it more readable. For the RL task, we use colored images such that the models may recognize the object which is controlled by the agent. The mean values per color channels are added to each objects state, as a simple encoding of appearance. We clarified this in the revision.
>
> [RL experiments]
> The main motivation of our RL experiments is to demonstrate planning based on an object-aware dynamics model learned on purely visual input, which to our knowledge has not been done in prior work. Wang et al. use GNNs very differently from us, by employing them in a model-free policy network. Sanchez-Gonzalez et al., like us, use GNNs as a dynamics model for planning, but assume access to the ground truth states as opposed to inferring them from images.
>
> [Realistic Rollouts]
> We find that STOVE significantly improves upon prior work in that it predicts physical behavior across long timeframes, instead of stopping or teleporting objects. We quantify this in the revision by plotting the conservation of kinetic energy in the rollouts, which STOVE achieves up to at least 100,000 steps, while DDPAE and SQAIR break down after less than 100. See (1) of our top level comment and the animated GIFs in our anonymized GitHub [1].
>
> [Ablations]
> In the revision, we provide results for three ablations (see (3) in our general comment and Table 1), including two with an ablated state representation. We did not explore AIR as an alternative object detector, since we chose SuPAIR for its faster training times. We do not claim, or even expect, that AIR would perform worse.
>
> [Steenkiste et al.]
> For a visual evaluation, please compare our animated rollouts [1] with the ones presented by Steenkiste et al. [2, very bottom]. We find that STOVE more accurately captures object permanence and energy conservation. We decided against a quantitative comparison due to qualitative differences:
> (a) R-NEM requires around 10 given observations before the iterative inference procedure converges to a good segmentation,
> (b) it does not explicitly model object positions, and
> (c) it requires noisy input to avoid local minima.
> We have instead added DDPAE as a baseline. See (2) in our top level comment.
>
> [1] https://github.com/ICLR20/STOVE
> [2] https://sites.google.com/view/r-nem-gifs/
>
> [RL Performance]
> The performance of MCTS+STOVE was very close to the performance of MCTS on the ground truth environment. This indicates that the weak point of the agent was not the model (STOVE), but rather the planner, and that more thorough planning would allow it to match PPO's performance. Since the goal of our RL experiments was to highlight the applicability and sample efficiency of our model in the RL domain, we opted for an off-the-shelf planner instead of tuning for final performance.
>
> [Suggested Related Work]
> Thank you for the references, we have added them.
>
> [Reuse of Dynamics Model]
> Previous models, such as SQAIR and DDPAE, use an inference distribution $q(z_t | x_t, z_{t-1})$ which is entirely separate from the generative dynamics model $p(z_t | z_{t-1})$. We argue that this is wasteful, as much of the knowledge captured by the generative dynamics model is also relevant for the inference network. We therefore reuse it in our formulation of the inference network (Eq. 2), saving model parameters and regularizing training. We explore the benefits of this in one of the new ablations ("double dynamics").
>
> [Failure Modes]
> The main failure mode is that the inductive bias in the image model is insufficient to reliably detect objects. See Stelzner et al. for a discussion of noisy backgrounds in SuPAIR. In addition, our matching procedure assumes that objects move continuously.
>
> [Occlusion]
> Occlusion is explicitly modelled in SuPAIR: If objects overlap, the hidden parts of the occluded object are treated as unobserved, and therefore marginalized during the evaluation of the object appearances' likelihood.
>
> We hope that the changes made will address your concerns and look forward to further discussion.

---

> > ### Comment · AnonReviewer1 · 2019-11-12
> > **Reviewer (#1) response to author response**
> >
> > Thank you for your detailed response. My questions and comments are addressed and I think the revised version of the paper meets the bar for acceptance at ICLR.
> >
> > One minor note: In the revised version of the paper, you use “graph network” and “graph neural network” interchangeably — maybe you could consider consistently just using either one of the two terms to avoid potential confusion.

---

> > > ### Author Response · Authors · 2019-11-12
> > > **Response to updated review #1**
> > >
> > > Thank you for updating your review. We will make sure to stick to the term "graph neural network" in the camera-ready version.

---

> > ### Comment · AnonReviewer3 · 2019-11-14
> > **Reviewer #3 response to author response**
> >
> > Thank you for the clear response and updated document. I have updated my review in response as I now believe that the paper should be accepted.

---

> > > ### Author Response · Authors · 2019-11-14
> > > **Response to updated review #3**
> > >
> > > Thank you for updating your evaluation. We are glad that the revision addressed your concerns.

---

### Official Review · AnonReviewer3 · 2019-10-22
**Official Blind Review #3**

**Rating:** 6

**Review:**

In this paper the authors present a graph neural network for modeling the dynamics of objects in simple environments from video. The intuition of the presented system is that it first identifies the different objects from the image using Sum-Product Attend-Infer-Repeat (SuPAIR), which gives the objects positions and sizes. The system uses a “simple matching procedure” to map objects between frames, which allows for the system to extra the object’s velocities. Then a graph neural network is employed to model the dynamics of the particular environment (whether objects bounce, whether there are other forces at play like gravity, etc.). The authors present two environments (Billiards and Gravity) and two evaluations, one focused on predicting future states, and the second focused on using these predictions to play the game.

I think that this paper presents an interesting approach and I agree with the authors of the importance of developing approaches that allow AI to make good predictions of future environments. However, I’m not convinced of many of the technical details in the paper.

I am not certain whether I would classify this work as unsupervised learning. While it’s certainly true that there are no labels in the raw video, the object-finding can be understood as a preprocessing step after which the data is in fact in a fairly standard supervised learning framework. The authors use the term “self-supervised” in the first section, which I believe describes the work more clearly.

The primary technical contributions of the work appear to be the graph network, the experiments, and their results. While I would have preferred more detail on the graph network in an appendix, it’s acceptable to instead have access to the code. However, the experiments seem set up primarily to evaluate the system as a whole. For example, the inclusion of a supervised learning version of the system where the object’s positions are given exactly sheds light on the quality of SuPAIR. However, SuPAIR is taken from prior work. I would have thought that an entirely different approach, like that used by Ha and Schmidhuber in their World Models paper would have been more appropriate as a comparison as it represents an alternate approach entirely.

There is a repeated claim made in the paper that the system presents output that is “convincing” and “realistic” over hundreds of time steps. There is no clear definition given for what this means. Figure 1 only presents pixel and positional error for 80 frames, and the error appears to go pretty large (~15%) after only forty frames. The results presented in Figure 4 suggests a much larger timescale, but it’s unclear the quality of the output predictions from it. Some clarity on this or scaling back the claims would improve the paper.

In terms of related work Guzdial and Riedl’s 2017 “Game Engine Learning from Gameplay Video” appear to use a very similar approach (but with OpenCV instead of SuPAIR and search instead of a graph network) as does Ersen and Sariel’s 2015 “Learning behaviors of and interactions among objects through spatio–temporal reasoning”. These approaches also function over much more complex environments with variable numbers of objects. It would be helpful for the authors to continue adding some discussion of this and related papers.

---

Edit: In response to the author's changes I have increased my rating to a weak accept. This is in large part due to Figure 4, which provides a great deal of additional support to the author's claims and clarity on the technical value of the results.

**Experience Assessment:**

I have published in this field for several years.

**Review Assessment: Checking Correctness Of Derivations And Theory:**

N/A

**Review Assessment: Checking Correctness Of Experiments:**

I carefully checked the experiments.

**Review Assessment: Thoroughness In Paper Reading:**

I read the paper thoroughly.

---

> ### Author Response · Authors · 2019-11-08
> **Response to Reviewer 3**
>
> Dear Reviewer 3,
>
> thank you for your valuable feedback.
> Below, we give a detailed response to your questions and comments.
>
> [Realistic Rollouts]
> We have quantified the notion of realistic rollouts by adding a plot of the kinetic energy in the billiards ball system across prediction timesteps. This energy should be conserved, as collisions are fully elastic and energies thus remain constant in the training data. For STOVE, the mean energy remains constant even over extremely long timeframes (we checked up to 100,000 steps), whereas for the baselines, it quickly diverges (after less than 100 steps). While in chaotic systems like the billiards environment, model predictions will necessarily differ from the ground truth after a number of timesteps, it is a desirable property of STOVE to continue to exhibit physical behavior. In contrast, all baselines predict overlapping, stopping, or teleporting objects after a short period. This can be observed visually in our animated GIFs [1].
>
> [1] https://github.com/ICLR20/STOVE
>
> [Unsupervised Learning]
> We agree that 'self-supervised' is a good term for STOVE. However, we do not view the end-to-end learning approach of STOVE as equivalent to decomposing the task into two distinct steps, one for feature extraction and one for supervised prediction. Kosiorek et al. (SQAIR) have shown that training dynamics and recognition models jointly can significantly improve object detection performance through the incorporation of a temporal consistency bias. We therefore believe that maintaining this coupling is a valuable feature of STOVE. In any case, the successive training of SuPAIR and dynamics model is more brittle and raises the need for additional auxiliary losses (as in Watters et al. (2017)), such as a carefully tuned discounted rollout error.
>
> [Contribution]
> As requested, we have added detailed information on the graph neural network and other components of STOVE to the appendix. We disagree with the assessment that our paper's main contribution is the graph network architecture. The benefits of relational architectures for multi-object dynamics tasks have previously been demonstrated, e.g. by Battaglia et al. (2016) and Watters et al. (2017). What has not been done before is to employ them in a setting in which state information is entirely latent, and only raw video is available. Our main contributions are to show how to do this (structured latent space, reuse of the dynamics model, joint variational inference), and to demonstrate that this enables predictions of comparable quality to the supervised setting with observed states. This comparison does not merely evaluate SuPAIR, but rather the techniques we proposed for connecting image and dynamics models.
>
> [Ha & Schmidhuber]
> We compare to VRNN, which belongs to the same class of model as the one Ha & Schmidhuber propose. Both encode input images via a VAE, and model the dynamics of the latent state via an RNN. It has been repeatedly demonstrated in the literature that models with object-factorized state representations such as STOVE outperform models with unstructured states, and our results support this, too. See e.g. the papers on SQAIR (Kosiorek et al., (2018)), and DDPAE (Hsieh et al., (2018)). We therefore deem a comparison to VRNN as a representative of unstructured models sufficient.
>
> [Diverse Number of Objects]
> Even though we did not explore this in this paper, one of the main appeals of both GNNs and AIR-based models is the ability to handle a variable number of objects. This is enabled by the GNNs focus on pairwise interactions. STOVE can thus be easily extended to handle a variable number of objects. As an ad-hoc demonstration, we provide an animated rollout with 6 objects on our GitHub [1].
>
> [Game Engine Learning]
> Both Ersen & Sariel and Guzdial & Riedl share our motivation of learning the rules of games from video, we have therefore added the references. However, they explore a very different setting, since they assume access to a curated set of sprites to handle object detection, and use logical rules instead of continuous dynamics to model interactions. We find it misleading to credit these works with being able to handle more complex visual environments, as the a-priori knowledge of pixel-perfect object appearances trivializes the detection task. The goal of the field of representation learning, including AIR and all of its derivatives, is to extract meaningful, potentially discrete information from noisy and continuous input data without relying on domain specific knowledge. While hand-engineered approaches to object detection would certainly work on the domains we considered here, the techniques we present in this paper generalize to different image models and different environments. It is our hope that models like ours will make it possible to apply logical reasoning to domains where it was previously impossible, because of their continuous and noisy nature, and the absence of domain-specific knowledge.

---

### Official Review · AnonReviewer2 · 2019-10-23
**Official Blind Review #2**

**Rating:** 6

**Review:**

This paper presents STOVE, an object-centric structured model for predicting the dynamics of interacting objects. It extends SuPAIR, a probabilistic deep model based on Sum-Product Networks, towards modeling multi-object interactions in video sequences. Compared to prior work, the model uses graph neural networks for learning the transition dynamics and reuses the dynamics model for the state-space inference model, further regularising the learning process. The approach has been tested on simple multi-body physics tasks and performs well compared to other unsupervised and supervised baselines. Additionally, an action-conditional version of STOVE was tested on a visual MPC task (using MCTS for planning) and was shown to learn significantly faster compared to model-free baselines.

The paper is well written and clearly motivated but comes across as an incremental improvement on top of prior work. Here are a few comments:
1. The idea of reusing the dynamics model for inference is neat as it helps to regularise the learning process and remove the costly double recurrence, potentially speeding up learning. It would be great if this could be evaluated experimentally via an ablation study — this can be done by using two separate instances of the transition model with separate weights.
2. A keys step that allows to reconcile the transition model and the object detection network is the matching process. Currently, this is done via choosing the pair with the least position and velocity difference between subsequent time steps. This could give erroneous results in the case of object interactions when objects are fairly close to each other (or colliding). A potentially better way could be to additionally use the content/latent codes for this matching process — as long as the object’s appearance stays similar these can provide good signal that disambiguates different objects.
3. The experiments presented in the paper are quite simplistic visually — it is not clear if this approach can generalise to more complicated visual settings. Additionally, it would be good to see further comparisons and ablations that quantifies the effect of the different components — e.g. comparing to a combination of image model + black-box MLP dynamics model can quantify the effect of the graph neural network. These results can add further strength to the paper.

Overall, the approach presented in the paper is a bit incremental and the experiments are somewhat simplistic. Further comparisons and ablation experiments can significantly	strengthen the paper. I would suggest a borderline accept.

**Experience Assessment:**

I have published one or two papers in this area.

**Review Assessment: Checking Correctness Of Derivations And Theory:**

I assessed the sensibility of the derivations and theory.

**Review Assessment: Checking Correctness Of Experiments:**

I assessed the sensibility of the experiments.

**Review Assessment: Thoroughness In Paper Reading:**

I read the paper at least twice and used my best judgement in assessing the paper.

---

> ### Author Response · Authors · 2019-11-08
> **Response to Reviewer 2**
>
> Dear Reviewer 2,
>
> thank you for your valuable feedback.
> Below, we give a detailed response to your questions and comments.
> Please also see the changes to the manuscript outlined in our top level comment.
>
> [Ablations]
> We have added results for three different ablations of STOVE, including the suggested one in which two separate dynamics nets are used for generation and inference, demonstrating the value of reusing the dynamics net. Please see (3) in our general comment and Table 1 in the manuscript. We have chosen not to explore black-box MLPs as dynamics models, as the benefits of graph neural networks for multi-object dynamics tasks are well documented in the literature, see e.g. Battaglia et al. (2016) and Watters et al. (2017). We therefore do not believe this to be a crucial baseline.
>
> [Appearance-Based Matching]
> We agree, and have tried matching procedures which involve object appearance encodings. However, one of the main features of SuPAIR in contrast to AIR is that it does not necessitate a latent encoding of the object appearance. This means that an encoder network would have to be 'tacked on' to the model in order to allow for appearance based matching, as mentioned in Section 2.4. We did not find this necessary, since for the settings we considered, STOVE precisely inferred object centers with a mean error of less than 1/3 of a pixel, which suffices even during collisions or in scenarios with partial overlap. We therefore leave the exploration of appearance-based matching to future work.
>
> [Visual Complexity]
> The visual complexity of scenes and robustness of SuPAIR with respect to visual noise has been explored by Stelzner et al.. We expect that these results translate to STOVE, i.e., that STOVE is able to handle background noise better than AIR (and, by extension, DDPAE and SQAIR). Figure 5 in the appendix shows that we are able to model scenes of differently shaped object sprites. However, we did not focus on this in this paper, as its main contributions are the techniques presented to combine image and dynamics models, as opposed to the performance of the specific image model used. Due to the compositional nature of STOVE, more sophisticated image models may easily be plugged in in place of SuPAIR. Finally, we note that the complexity of the experiments is in line with previous work (DDPAE, R-NEM). We choose to extend them by exploring the RL domain, which brings additional challenges, such as dynamics depending on object identities and actions.
>
> [Meaningful Improvement]
> Please see (1) of our top level comment, as we believe the energy conservation plot clearly demonstrates the stark performance improvements achieved with STOVE over prior work. While previous approaches break down after less than 100 frames of rollout, STOVE predicts trajectories with constant mean energy trajectories for 100,000 frames or more. Additionally, DDPAE and SQAIR predict overlapping, stopping, or teleporting objects after a short period. Apart from the added conservation plot, this is also apparent from the animated GIFs in our anonymized GitHub [1].
>
> [1] https://github.com/ICLR20/STOVE

---

### Author Response · Authors · 2019-11-08
**Revision**

We thank the reviewers for their valuable feedback and have revised the manuscript accordingly.

The main changes are:

1) We quantify the notion of 'realistic' rollouts by plotting the kinetic energy of rollouts on the billiards task (Fig. 4). This energy should be conserved, as collisions are fully elastic and no friction is applied. We find that the energy in STOVE's rollouts remains constant over very long timeframes (we checked up to 100,000 steps), whereas it quickly diverges for the baselines (SQAIR and DDPAE) after less than 100 steps. Additionally, all baselines predict overlapping, stopping, or teleporting objects after a short period. This stark difference in quality can be observed visually in our animated GIFs [1].

2) We have added DDPAE as a baseline for the video prediction tasks. According to its authors, DDPAEs is capable of handling complex interactions on the billiards task. In our experiments, it performs better than SQAIR, but significantly worse than STOVE.

3) We have added results for three of the suggested ablations to table 1. They are:
   a) Double Dynamics Networks (two separate dynamics nets for inference and generation),
   b) No Velocity (no explicit modelling of the velocity within the state),
   c) No Latents (no unstructured latent variables in the dynamics state, only positions and velocities).
We find that they all perform consistently worse than full STOVE, demonstrating each component's value.

4) We fixed a bug in our RL environment and adjusted the results accordingly. PPO now converges slightly faster (4M instead of 5M steps), but all high-level observations remain the same.

5) We have improved the clarity of the writing.

6) We have added a detailed description of the model architectures, hyperparameters, and baselines to the appendix.

7) We have included the suggested additional references.

We hope these changes address the concerns expressed by the reviewers and look forward to further discussion.

[1] github.com/ICLR20/STOVE

---

### Decision · Program_Chairs · 2019-12-19

**Decision:**

Accept (Poster)

**Comment:**

The paper presents a method for modeling videos with object-centric structured representations. The paper is well written and clearly motivated. Using a Graph Neural Network for modeling latent physics is a sensible idea and can be beneficial for planning/control. Experimental results show improved performance over the baselines. After the rebuttal, many questions/concerns from the reviewers were addressed, and all reviewers recommend weak acceptance.